# Renal Carcinoma and Angiogenesis: Therapeutic Target and Biomarkers of Response in Current Therapies

**DOI:** 10.3390/cancers14246167

**Published:** 2022-12-14

**Authors:** Zoé Guillaume, Marie Auvray, Yann Vano, Stéphane Oudard, Dominique Helley, Laetitia Mauge

**Affiliations:** 1Université Paris Cité, Inserm, PARCC, F-75015 Paris, France; 2Service d’Oncologie Médicale, AP-HP, Hôpital Européen Georges-Pompidou, F-75015 Paris, France; 3Service d’Hématologie Biologique, AP-HP, Hôpital Européen Georges-Pompidou, F-75015 Paris, France

**Keywords:** angiogenesis, renal cell carcinoma, biomarker, VEGF-A, endothelial cell, immunotherapy

## Abstract

**Simple Summary:**

The treatment of renal cancer is currently based on the use of antiangiogenic drugs targeting the VEGF-A pathway and/or immunotherapy targeting immune checkpoint inhibitors. Despite combined therapies being approved as first-line treatments, all patients will not benefit from them. We highlight here the role of tumour angiogenesis in renal cancer which makes angiogenesis-related markers good candidates to predict response to treatments including immunotherapies. Less data is available in this field for recently combined treatments. A combination of angiogenesis-related biomarkers with markers of other processes would be relevant to progress in the aim of personalized treatment.

**Abstract:**

Due to the aberrant hypervascularization and the high immune infiltration of renal tumours, current therapeutic regimens of renal cell carcinoma (RCC) target angiogenic or immunosuppressive pathways or both. Tumour angiogenesis plays an essential role in tumour growth and immunosuppression. Indeed, the aberrant vasculature promotes hypoxia and can also exert immunosuppressive functions. In addition, pro-angiogenic factors, including VEGF-A, have an immunosuppressive action on immune cells. Despite the progress of treatments in RCC, there are still non responders or acquired resistance. Currently, no biomarkers are used in clinical practice to guide the choice between the different available treatments. Considering the role of angiogenesis in RCC, angiogenesis-related markers are interesting candidates. They have been studied in the response to antiangiogenic drugs (AA) and show interest in predicting the response. They have been less studied in immunotherapy alone or combined with AA. In this review, we will discuss the role of angiogenesis in tumour growth and immune escape and the place of angiogenesis-targeted biomarkers to predict response to current therapies in RCC.

## 1. Introduction

Renal cell carcinoma (RCC) represents 3 to 5% of all cancers in the world with an increasing incidence of approximately 400,000 cases in 2018 [1]. The predominant histological form is clear cell renal cell carcinoma (ccRCC) (>75%) and even today, the prognosis of this disease remains poor as 30% of patients have metastatic disease at diagnosis and the 5-year survival is estimated at 12% [2].

The development of new research tools has led to a better understanding of the biological and molecular mechanisms underlying the development of these cancers. Angiogenesis is required for tumour growth [3]. Indeed, tumour vessels bring oxygen and nutrients for tumour cells to survive and proliferate. Angiogenesis is also involved in tumour evasion from the immune system either directly or indirectly [4]. First, tumour vessels control immune cell infiltration. Second, abnormal tumour vascularization can promote hypoxia and many proangiogenic factors exert immunosuppressive functions. Angiogenesis is an even more important target in renal cancers because they present an aberrant angiogenesis and are resistant to chemo- and radiotherapy. Indeed, a very frequent mutation of the tumour suppressor gene *VHL* in ccRCC (>80% of cases) dysregulates hypoxia-inducing factor (HIF) inducing overexpression of vascular endothelial growth factor A (VEGF-A) and platelet-derived growth factor (PDGF) leading to tumour growth [5,6]. Tyrosine kinase inhibitors (TKIs) targeting the VEGF pathway have thus replaced IL-2 and IFN-α since 2005 [7,8]. 

Several years ago, the therapeutic arsenal was enriched with a new class of immunotherapy, immune checkpoint inhibitors (ICIs) targeting PD-1 or CTLA-4 [9,10]. Indeed, renal cancer is also one of the most immunogenic cancers with a tumour microenvironment (TME) characterized by an infiltration of various immune cells with an immunosuppressive phenotype [11]. Considering the link between angiogenesis and anti-tumour immunity, there is a strong rationale to combine ICI (anti-PD-1) with anti-angiogenic TKI [12]. Four combinations are currently approved for first-line treatment in m ccRCC (3 TKI-ICI-axitinib plus pembrolizumab, cabozantinib plus nivolumab and lenvatinib plus pembrolizumab—and 1 ICI-ICI-nivolumab plus ipilimumab) [13]. Despite ORRs ranging from 55% to 70% and a gain in PFS and OS, there are still patients who do not respond to these treatments, durable responses remain extremely rare, and progression is almost systematic. Biomarkers are therefore essential to target patients who could benefit from these treatments or to anticipate the occurrence of secondary resistance. 

This review first describes angiogenesis and its role in tumour growth, particularly in renal cancer. To understand the importance of angiogenesis-related biomarkers to predict the response not only for anti-angiogenic TKI treatments, angiogenesis involvement in tumour immune escape is reported. We then review the interest in angiogenesis-related biomarkers to predict the response to TKI and recently available treatments. Whereas they have been extensively explored in the era of TKI treatments, biomarkers studied since the use of ICI were mostly related to immunity. They have proven to be useful in other cancers, but do not appear to predict responses in RCC [14,15]. The role of angiogenesis in predicting response to ICI or combination therapies remains to be clearly determined. 

## 2. Methods

Angiogenesis being already well documented, we searched for English reviews from 2019 onwards as well as articles studying angiogenesis specifically in renal cancer. Concerning the part of the review focusing on angiogenesis biomarkers, the following literature search strategy was applied. We performed a systematic search on the PubMed bibliometric database including English-language articles published up to July 2022 reporting data relevant to biomarkers of current treatments, i.e., AA, ICI, or both in RCC patients. The following keywords were used: “renal carcinoma” “biomarker” “angiogenesis” or “VEGF”, « vessel », “immunotherapy”, and “combination”. Supplemental manual searches were conducted from congresses (2020–2022) of greatest relevance (the Annual Meeting of the American Society of Clinical Oncology [ASCO]; the ASCO Genitourinary Cancers Symposium [ASCO GU]; the Annual Meeting of the European Society for Medical Oncology [ESMO]). Some additional key articles that were published after the bibliometric search were identified by the authors and included. The objective of this part of the review was not to be exhaustive on anti-angiogenic TKI biomarkers, already highly described [16,17], but to report the mainly observed results and review the place of these angiogenesis-related biomarkers to predict response to recently used TKI, ICI, and combinations used in ccRCC. 

## 3. Angiogenesis in Renal Cancer

### 3.1. Angiogenesis

Angiogenesis is one of the processes by which existing blood vessels form new ones [18]. This process is physiological and essential during embryonic life. It becomes quiescent during adult life except in certain conditions such as wound healing, ovarian cycle, or pregnancy. However, angiogenesis can be associated with pathological phenomena such as the development of cancers. Indeed, a tumour would not be able to measure more than 1 mm^2^ without the establishment of a vascular network allowing the supply of oxygen and nutrients [3].

Angiogenesis involves the development, migration, and proliferation of endothelial cells (EC) [18,19]. It is regulated by numerous pro- and anti-angiogenic factors. The main steps of angiogenesis are presented in Figure 1.

Other processes of new vessel formation can be observed in tumours, such as vasculogenesis, co-option, or vascular mimicry [18]. Vasculogenesis relies on the recruitment of bone marrow-derived and/or vascular wall resident endothelial progenitor cells (EPCs) towards the tumour that differentiate into mature endothelial cells. Tumour cells can co-opt pre-existing vessels. Aggressive tumour cells can also form vessel-like structures without the contribution of endothelial cells. This mechanism, called vascular mimicry, has been seen in many cancers. Endothelial-like tumour cells express then endothelial markers such as CD31 [20,21]. This alternative neovascularization process adopted by the tumour may cope with the treatment and overcome the hypoxic environment [21].

### 3.2. Place of VEGF-A in Cancer Angiogenesis

#### 3.2.1. Angiogenic Switch

VEGF-A is one of the most important factors in angiogenesis. It belongs to a family of 7 members: Placental Growth Factor (PlGF) 1 and 2, VEGF-B, VEGF-C, VEGF-D, and VEGF-E (a viral gene). While VEGF-A is a key regulator of angiogenesis, VEGF C, and D regulate lymphangiogenesis [22]. The effect of VEGF-A on ECs is mediated by intracellular signaling after binding to its receptors: VEGFR-1 and VEGFR-2. VEGFR-2 is the main signal transducer in angiogenesis, whereas VEGFR-1 plays a negative role in angiogenesis by maintaining an appropriate level of VEGFR-2 activation [23]. Signaling of VEGF-A through VEGFR-2 induces EC proliferation, migration, and increases vascular permeability and mobilization of endothelial progenitor cells (EPCs) [24]. Through VEGFR-1 binding, VEGF-A favors monocyte migration, allows the recruitment of EPCs, and increases the adhesive properties of NK cells. VEGFR-3, a receptor for VEGF-C is necessary for the formation of the blood vasculature during early embryogenesis, but later becomes a key regulator of lymphangiogenesis. Furthermore, NRP-1 and NRP-2 (neuropilins), better known for their role in neuronal development, are co-receptors for VEGFR-1 and 2 and increase the ligand affinity for these receptors suggesting a potential role in angiogenesis [24].

Hypoxia is a major factor driving angiogenesis. It decreases the degradation of the transcription factors HIF-α by ubiquitination and finally induces the expression of proangiogenic factors, among them VEGF-A (Figure 2). Contrary to HIF-1α which is a ccRCC suppressor, HIF-2α is a ccRCC oncoprotein and therefore an interesting therapeutic target already evaluated in several phase 2 that showed anti-tumour activity [25,26]. Phase 3 studies are ongoing to evaluate HIF-2α inhibitors as a single agent or in combination with immunotherapy (NCT03634540, NCT04736706). HIF-2α activates various genes encoding molecules that probably have a causal role in the development of ccRCC including angiogenic growth factors VEGF-A, PDGFB, and SDF-1 [27].

VEGF-A expression is also upregulated by growth factors, cytokines, and hormones such as estrogens [28]. In cancer, VEGF-A can be secreted by various types of cells following hypoxia: tumour cells mainly, but also fibroblasts, myeloid-derived stem cells, or NK cells [29]. VEGF-A production by tumour cells can also be due to oncogenic events. The disruption of the balance in favour of proangiogenic factors is essential for tumour growth and is called the “angiogenic switch”.

#### 3.2.2. VHL and PBRM1 Inactivation in RCC

Clear cell RCC is usually associated with a mutation in the *VHL* gene. Mutations in *VHL* are either responsible for a loss of function or hypermethylation of the promoter making it non-functional. They can be somatic or secondary to a rare germline mutation leading to Von Hipple-Lindau disease which is responsible for a predisposition to certain types of hypervascular carcinoma such as ccRCC. Loss of function of *VHL* leads to an accumulation of HIF-α and mimics or aggravates a hypoxic situation in the tumour, leading to an increase in PI3-K/PKB/mTOR signalling and tumour progression [30,31]. 

*VHL* inactivation is usually the initiating event in the development of ccRCC. However, it is not sufficient to cause the occurrence of ccRCC, other cooperating genetic events are required, such as the loss of function of the tumour suppressor gene *PBRM1*. *PBMR1* is the second most common gene mutated in ccRCC after *VHL* mutations. This gene encodes a component of a multiprotein SWI/SNF complex that regulates the position of nucleosomes in the genome [32]. *PBRM1* knockdown increases the proliferation and migration of kidney cancer cell lines [33]. In addition, the loss of *PBRM1* amplifies the transcriptional effects of HIF-1 and STAT3 caused by the loss of *VHL* [34]. Clinical studies have observed that alterations in *PBRM1* are associated with increased expression of angiogenesis genes in ccRCC [35,36]. It, therefore, has an indirect role in tumour angiogenesis and is currently being studied extensively in ccRCC [36,37]. VHL-/- ccRCC tumours and cell lines produce both HIF-1α and HIF-2α or HIF-2α alone [38].

#### 3.2.3. VEGF-A Promotes Immunosuppression

Tumour-induced angiogenesis is not only essential for tumour growth but also contributes to immune evasion through the induction of a highly immunosuppressive tumour microenvironment (TME). Excess of proangiogenic factors favours immunosuppression, both through their effects on tumour vasculature and immune cells (Figure 3).

Mature vessels control many processes: vascular tone, permeability, inflammation, and coagulation [39]. Due to the imbalance in favour of proangiogenic factors, tumour vessels, unlike normal vessels, present abnormal characteristics. They are often immature, dilated, tortuous, disorganized, and therefore not fully functional [40]. They are also leaky, with poorly developed cell–cell contacts, and they display reduced pericyte coverage. Such aberrant tumour vessels can lead to a decrease in the supply of nutrients and oxygen and thus promote hypoxia in the TME which worsens tumour angiogenesis, immunosuppressive phenotype, and select resistant clones [41]). The increased permeability also facilitates tumour cell evasion. In the same tumour, vascularization can be heterogeneous with hypervascularized areas and others less. This also impacts the response to treatment.

Moreover, proangiogenic factors promote the creation of a selective immune-cell barrier [4,42]. Quiescent ECs control inflammation and limit access of leucocytes to the tissue [39]. When activated, ECs allow the recruitment and tissue infiltration of immune cells, particularly lymphocytes, to inflammatory sites through the expression of cell adhesion molecules such as ICAM-1 and VCAM-1. VEGF-A and bFGF, for example, decrease TNF-α induction of CAM expression by EC by blocking NFkB and inducing NO production [43]. Expression of molecules favouring the selective infiltration of immunosuppressive cells has also been reported, such as CLEVER-1, which promotes selective infiltration of regulatory T cells [44]. Fas-L expression induced by VEGF-A promotes a selective entry of regulatory T cells in the tumours owing to their higher expression of the anti-apoptotic factor c-FLIP, contrary to cytotoxic T cells [45]. It is interesting to notice that IFN-γ and TNF-α, important mediators of antitumour immune responses promoting PD-L1 and PD-L2 expression, inhibit tumour angiogenesis [46]. Under pro-inflammatory signals, EC can also present an immunosuppressive phenotype through the expression of immune checkpoints PD-L1 and PD-L2 or the production of immunosuppressive cytokines. Despite preclinical data in favour of an immunosuppressive role, the exact immunomodulatory role of ECs expressing PD-L1 or PD-L2 in tumours remains under investigation.

Proangiogenic factors involved in tumour angiogenesis can also promote immunosuppression by direct effects on immune cells. VEGF-A immunosuppressive role has been well described [47,48,49,50]. It inhibits the maturation of dendritic cells, with or without the help of NRP1, thus altering the presentation of tumour antigens. It favours the recruitment and proliferation of LTreg or other immunosuppressive cells such as MDSCs (myeloid-derived suppressor cells). Through its binding to VEGFR-2 expressed by CD8+ T cell, it increases the exhaustion of LT by promoting the expression of inhibitory checkpoint molecules such as PD-1, CTLA-4, LAG-3, and TIM-3. Similar immunosuppressive effects of other proangiogenic factors including hepatocyte growth factor (HGF), PDGF, and angiopoietins have also been reported [51].

### 3.3. Tumour Angiogenesis Contributes to Drug Resistance in RCC

#### 3.3.1. Resistance to TKI

Mechanisms of resistance to AA and ICI rely on multiple pathways. The role of angiogenesis in tumour resistance to treatment has been largely reported for anti-angiogenic TKI used in monotherapy, such as sunitinib [52,53]. These mechanisms are either already present and explain primary resistance or can be induced secondary to TKI treatment. 

Hypoxia-driven activation of other pro-angiogenic pathways inducing resistance has been reported. PDGFR, MET, AXL, and FGFR have been shown to play a role in resistance to VEGFR TKI [54,55]. Upregulation of PlGF (Placental growth factor) and angiopoietin 2 have also been described in ccRCC patients developing resistance to TKI [52,53]. These observations have guided the development of TKI targeting complementary pathways such as cabozantinib which inhibits c-Met [56] and lenvatinib which inhibits other FGFR [57]. Hypoxia also promotes the secretion of chemoattractants such as SDF-1 which recruits bone-marrow derived pro-angiogenic inflammatory cells in the TME. 

Other mechanisms of resistance described for TKI include the metabolism adaptation of tumour cells to hypoxia, vascular co-option, epithelial-mesenchymal transition, lysosomal sequestration of the drug, epigenetic modification of histone protein and overexpression of PD-L1 [58,59]. Emerging evidence indicates that non-coding RNA such as micro RNAs (miRNA) or long non-coding RNAs (LncRNA) act as modulators of angiogenesis but not only and are involved in TKI resistance [60,61,62,63]. It might provide novel clinical markers and therapeutical targets for ccRCC patients, but due to their complex role and few data in ccRCC patients, at least for lncRNA, they will not be discussed further in this review. 

#### 3.3.2. Resistance to ICI

Inversely, the role of tumour angiogenesis in the resistance to ICI has been poorly studied. Resistance to ICI relies on an alteration of one of the following steps: antigen presentation and T cell priming, T cell infiltration in the tumour, and cytotoxic T cell activity in the tumour [58]. First, tumour vessels regulate T cell infiltration from peripheral blood into the tumour. Second, as explained previously, they can inactivate T cells by expressing checkpoint inhibitors or secreting immunosuppressive cytokines, like immune and tumour cells. Third, tumour hypoxia confers tumour cells resistance to both TKI and ICI and is usually associated with a bad prognosis [58]. Indeed, hypoxia promotes immunosuppression and tumour aggressiveness to survive in hypoxic areas. As explained previously, the importance and functionality of tumour angiogenesis regulate hypoxia in the tumour and are thus involved in the resistance to treatment. Angiogenesis biomarkers should thus be included in the biomarker studies performed in patients treated with immunotherapy. 

## 4. Angiogenesis Related Biomarkers in ccRCC 

Angiogenesis-related biomarkers have been extensively studied to identify a predictive marker of the first TKI treatments used in ccRCC, mostly sunitinib [52,64]. Considering the role of tumour angiogenesis in immunity, the study of angiogenesis-related biomarkers is relevant in ccRCC patients treated with immunotherapy or combined therapy. The main results observed with TKI will be summarized here and completed with observations reported in patients treated with ICI combined or not with TKI. 

### 4.1. Angiogenesis Blood Biomarkers

#### 4.1.1. Circulating Markers Related to VEGF-A Pathway

As a target of antiangiogenics, studies focused mainly on the VEGF-A pathway including VEGF-A and the soluble forms of its receptors. The baseline level of VEGF-A is often described as a prognostic factor in metastatic RCC (m-ccRCC): a high level of VEGF-A at diagnosis reflects an aggressive tumour [29,65]. It is associated with a shorter progression-free survival (PFS) and overall survival (OS) in patients with RCC treated with sunitinib and sorafenib [65,66,67,68,69,70,71] (Figure 4). VEGF-A level has also been studied in patients treated with everolimus, an mTOR inhibitor with an antiangiogenic activity, and discordant results were observed: higher levels associated with shorter survival [31] or no association [72]. A recent study evaluating lenvatinib plus everolimus versus lenvatinib alone or everolimus alone, included VEGF-A in a composite biomarker score (CBS). For the PFS score the markers studied were HGF, MIG/CXCL9, IL-18BP, IL-18, Ang-2, and for OS TIMP-1, M-CSF, IL-18BP, Ang-2, VEGF-A. Patients in high CBS score groups (with high VEGF-A levels) appeared to have improved PFS and OS with lenvatinib + everolimus combination therapy compared with everolimus monotherapy [73]. Concerning VEGF-A soluble receptors, sVEGFR-2 has been more studied than sVEGFR-1 in RCC. Low sVEGFR-2 baseline level was associated with a worse clinical outcome in m-ccRCC patients treated with sunitinib [70] whereas other studies reported no correlation, whatever TKI or everolimus [13,74,75]. Conversely, high levels of sVEGFR-1 were associated with shorter PFS and OS [70].

The study of blood biomarkers also allowed us to monitor the evolution of these markers over time according to clinical outcomes. VEGF-A level usually increases in patients with RCC treated with TKI or everolimus [76]. In addition, a higher increase of VEGF-A has been observed in the non-responders to sunitinib [74,76] and thus a significantly lower PFS than patients with a lower increase of VEGF-A levels after treatment (median PFS of 134 days vs. 367 days, *p* = 0.010, HR = 0.2 [95% CI = 0.059–0.68]. However, at the time of progression, patients who initially responded to sunitinib did not have increased VEGF-A levels [76]. Concerning soluble VEGF-A receptors, the decrease of sVEGFR-2 level is well described in m-ccRCC treated with antiangiogenic [77], but its association with clinical outcomes is not systematically observed. In patients with m-ccRCC treated with sunitinib, a significantly greater decrease in sVEGFR-2 levels was observed in patients with objective tumour response [70,74]. Soluble VEGFR-1 also decreases with sunitinib, but, conversely, the smaller the decrease in sVEGFR-1 during cycle 1, the longer the OS [70]. VE-cadherin is the major cell–cell adhesion molecule to maintain ECs adherent to each other. It is a direct target for sunitinib which inhibits its VEGF-induced phosphorylation and cleavage on the endothelial monolayer. Polena et al. have shown that, in RCC patients, a decrease in sVE level after four weeks of treatment discriminates the responders vs. non-responders to sunitinib, but not to bevacizumab [78].

Because of the central role of VEGF-A in tumour growth and immunosuppression, these molecules have also been studied in patients treated with immunotherapy. The BIONIKK trial, a phase 2 study, showed that high levels of sVEGFR-2 and VEGF-A were associated with a shorter PFS in patients treated with nivolumab for a m-ccCCRC [79]. In another trial evaluating atezolizumab in m-ccRCC patients, a biomarkers study showed that plasma VEGF-A decreased in responders but was stable in patients with stable disease or progressive disease [80]. Moreover, a recent study by Martini et al. evaluated blood markers in 52 m-ccRCC treatment-naive patients treated with axitinib and pembrolizumab [81]. Markers related to the VEGF-A pathway were studied: VEGF-A, sVEGFR-1, and sVEGFR-2. None of them was associated with treatment outcomes.

#### 4.1.2. Other Proangiogenic Pathways

The blood levels of several factors involved in angiogenesis have been investigated. A lower PDGF level at baseline was found in patients with clinical benefit on sunitinib (remission or stabilization) [76] and a lower Ang-2 level was associated with a better OS (motzer 2014). High levels of SDF-1 at baseline were associated with shorter PFS and OS in patients treated with sunitinib [70]. With regards to dynamic studies, increased SDF-1 in cycle 1 was significantly associated with response to treatment, whereas decreased PDGFB was greater in non-responders than responders. HGF exerts an angiogenic effect via c-Met by increasing the activation of the VEGF/VEGFR-2 pathway [82]. High baseline HGF levels were associated with shorter OS in patients receiving pazopanib [83,84]. Upregulation of these other proangiogenic pathways is involved in the resistance to TKI [52,53]. 

In patients treated with immunotherapy, either alone or in combination with TKI, few data are available on other proangiogenic pathways. In the BIONIKK trial, high levels of SDF-1 were associated with a shorter PFS in patients treated with nivolumab [79]. In patients treated with axitinib and pembrolizumab, blood levels of several angiogenesis-related factors were studied, among them Ang-1, Ang-2, cMET, HGF, IL8, MMP-9, SDF-1, and TGFβ [81]. None of them was associated with treatment outcomes.

#### 4.1.3. Circulating Endothelial Cells

Two types of circulating endothelial cells can be distinguished: mature circulating endothelial cells (CECs) derived from blood vessel wall upon vessel damage or high angiogenesis process [85,86,87] and circulating endothelial progenitor cells (EPCs) that contribute to neovascularization through vasculogenesis [88]. The increase of CEC after a few weeks of treatment is associated with longer PFS in m-ccRCC patients treated with sunitinib [75,89]. This could reflect vessel damage following a treatment with antiangiogenic TKI. Discordant results were observed concerning EPCs. Baseline levels > 2% were associated with shorter PFS and OS in m-ccRCC patients treated with sunitinib [90]. However, another study observed a significantly higher level of EPCs in responders to antiangiogenic TKI suggesting that tumours with active angiogenesis are more likely to respond to AA treatments [70]. Indeed, EPCs are involved in postnatal vasculogenesis, including tumour vascularization [91]. Methods used to quantify EPC and CEC are heterogeneous, as are their definitions. Added to the short storage time of the samples to study these cells, it is a limit for the use of these cells as biomarkers. To our knowledge, no studies including these biomarkers have been performed for patients treated with immunotherapy or combined treatments.

#### 4.1.4. Single Nucleotide Polymorphisms (SNP) 

Another interesting aspect to explain the difference in response to treatment is the search for polymorphisms in genes (germline variants) involved in angiogenesis. The susceptibility of individuals, particularly according to gene variations such as SNPs, has been studied as a biomarker of response and/or toxicity to treatments. These polymorphisms are studied on the DNA extracted from blood leucocytes. In patients with advanced renal carcinoma treated with pazopanib, *VEGF-A* polymorphisms were significantly associated with ORR, PFS, and OS [92,93]. The association of VEGF rs3025039 and VEGFR-2 rs2305948 genotype in 63 renal carcinoma patients treated with sunitinib was associated with shorter OS (*p* = 0.03) [94]. Polymorphisms of VEGFR-1 are also described as associated with response to sunitinib [95,96]. VEGFR-1 rs9582036 is associated with a poorer Objective Response Rate (ORR), PFS, and OS in m-ccRCC patients on sunitinib [97]. Other polymorphisms have been associated with response to sunitinib in *FGFR2* [52]. 

Due to the link between VEGF-A and HIF pathways, we also report here data on *HIF1A* polymorphisms. In patients with advanced RCC treated with pazopanib, *HIF1A* polymorphisms were significantly associated with ORR and PFS [92]. As well as *VHL* rs1642742 and rs1642743 which were associated with a poorer OS in m-ccRCC patients receiving first-line VEGFR-TKI [98]. 

We found no studies exploring polymorphisms in patients treated with immunotherapy or combinations in RCC.

### 4.2. Angiogenesis-Related Tumour Tissue Biomarkers 

Studying biomarkers in the tumour allows a specific reflect of the microenvironment. Tissue can be studied at a protein level (study of specific protein expression and vessel characterization), mRNA level (study of expression), or DNA level (study of mutations).

#### 4.2.1. Specific Protein Expression

VEGF-A overexpression has been associated with tumour progression and poor prognosis in several types of cancer like breast, prostate, or colorectal cancer [29]. For example, Garcia Dona et al. evaluated the expression of VEGF-A, VEGFR-1, VEGFR-2, and PDGFRB by immunohistochemistry in tumours from RCC patients treated with sunitinib sampled before treatment. High VEGF-A expression was associated with shorter OS (HR = 4.29, 95% CI 1.43–12.8, *p* = 0.0092). However, VEGF-A baseline expression in the tumour neither discriminates responders from non-responders, nor is associated with PFS in sunitinib-treated patients [99,100,101]. Another study evaluated VEGF-A expression according to its localization in the tumour center or margins. There was a significant correlation between tumour response and the difference in VEGF-A expression between the center and the margins of the tumour (*p* = 0.015). Indeed, the higher this difference was (stronger expression of VEGF in margins), the better the response [102].

Protein expression of VEGFR-1 -2 total or phosphorylated was also studied. Dornbusch et al. observed an association between high VEGFR-1 and 2 vessel expression at baseline and response to sunitinib (*p* = 0.048 and 0.010, respectively) [99]. High VEGFR-1 vessel expression was also associated with OS (HR: 0.449, 95% CI: 0.206–0.976). Another study reported an association of baseline expression of VEGFR-2, but not VEGFR-1, with PFS in m-ccRCC patients treated with sunitinib [103]. When studying VEGFR expression in tumours from patients after sunitinib treatment, Del Puerto-Nevado et al. reported a longer survival in patients with no phosphorylated VEGFR-2 on tumour vessels [104]. Results from VEGFR protein expression are not consistent among studies [100]. These discrepancies could be explained by the use of different clones of antibodies, the type of staining analysis taking into account the whole expression of the marker or specific endothelial expression, and the threshold of expression used to perform survival analyses. 

Considering the regulation of the VEGF-A pathway by the HIF axis, we report here the results on the expression of HIF-1α, HIF-2α and VHL studied by immunohistochemistry in patients with RCC treated with sunitinib [105]. Only high expression of HIF2A was associated with better clinical benefit (less PD) [OR = 0.11, 95% CI 0.02–0.75, *p* = 0.024 and longer survival (HR = 0.39, 95% CI 0.15–0.99, *p* = 0.048). In another study, high tumour HIF-1α and HIF-2α levels were both associated with sensitivity to sunitinib with better clinical response [106]. Conversely, HIF-1α was associated with a better PFS when it was poorly expressed (less than 50%) in another study [107].

#### 4.2.2. Tumour Vessel Characterization

Several studies have assessed vascular density through CD31 staining and its prognostic or predictive role in ccRCC. In general, increased tumour vascularization (e.g., increased microvessel density) and tumour expression of proangiogenic factors have been associated with advanced tumour stage and poor prognosis [29]. Bauman et al. showed an association of neovascularization (assessed based on the average CD105/CD31 expression ratio) with decreased overall survival (HR, 1.54 [95% CI, 1.06–2.23]) in 251 RCC patient tissues [108]. Moreover, two studies put into evidence a significant relationship between high vascular density (CD31 or CD34 staining) at baseline and response or a longer PFS and OS in patients treated with sunitinib [99,109]. These studies do not precise the phenotype or maturity of tumour vessels. In a study including 251 RCC patients treated with sunitinib, high pre-therapeutic Ang2 expression, and more strongly, combined high expression of both Ang2 and CD31, were associated with a high clinical benefit rate, but not with PFS or OS [110]. 

Other studies have characterized specific immunomodulatory functions of the tumour vasculature to assess its implication in the treatment of RCC. As an example, Inamura et al. demonstrate a positive association of B7-H3 expression in both tumour cells and the tumour vasculature with the density of tumour-infiltrating FOXP3+ cells [111]. B7-H3 is a member of the B7 family of immunoregulatory proteins, which includes PD-L1, and plays a critical role in the suppression of T-cell mediated antitumour immune responses, although conflicting evidence exists [112]. B7-H3 is also thought to control tumour aggressiveness in various types of cancer, including RCC [113]. While the physiological functions of B7-H3 remain elusive, B7-H3 is highly expressed in the tumour vessels of renal cancers, but not in the blood vessels of the corresponding normal tissue. In this study, high B7-H3 expression in the vasculature was independently associated with increased overall mortality (HR = 1.86, 95% CI = 1.05–3.45; *p* = 0.035). This association was even more relevant in the high FOXP3+ cell density group (HR = 4.86, 95% CI = 1.65–20.7; *p* = 0.0025). The same results were observed with B7-H3 expression by tumour cells. Immunosuppressive functions of tumour vessels have also been studied in patients receiving immunotherapy. Seeber et al. characterized IDO-1 expression in the tumour of RCC patients treated with nivolumab [114]. This negative immune-regulatory molecule was predominantly expressed in tumour ECs and was totally absent from tumour cells themselves. IDO-1 overexpression (>10%) could be detected more frequently in responders (100%, n = 6/6) compared to non-responders (33.3%, n = 3/9; *p* = 0.028), resulting in a better PFS during immunotherapy (IDO1 ≤ 10% vs. >10%, median: 3.5 vs. not estimated (NE) months, *p* = 0.01 by log-rank test). In addition, IDO-1 was positively correlated with CD8+ T cell expression (rs = 0.691, *p* = 0.006). Conversely, PD-L1 expression on tumour cells or tumour-infiltrating immune cells was irrespective of therapeutic response.

#### 4.2.3. mRNA Expression

Transcriptomic analyses give precise information on gene expression by quantifying the corresponding mRNA and identifying active genes within the tumour. Several studies focused on angiogenic genes. Others integrated them in larger molecular signatures that explore several complementary processes involved in tumour development and response to treatment. In accordance with the data of histochemical analyses, strong expression of angiogenic genes is usually associated with better outcomes in patients treated with TKI. One study assessed the mRNA expression of angiogenesis-related genes in ccRCC patients treated with sunitinib and showed that these genes were related to a better outcome [115]. On multivariate analysis, PDGFRB, VEGFR-1, VEGFR-2, and HIF2A expression were correlated with PFS and VEGFR-1, VEGFR-2, HIF1A et HIF2A expression with OS. Overall, VEGFR-2 expression showed the strongest association with all outcomes. In contrast to blood VEGF-A levels, lower VEGF-A expression in the tumour was associated with more progressive disease as best response *p* = 0.008; PFS HR: 1.166 95% CI 1.018–1.336 *p* = 0.027 and OS: HR:1.189 95% CI: 1.036–1.364 *p* = 0.014. High CD31 mRNA expression has also been reported as a good prognosis biomarker in ccRCC patients [116].

The study of molecular signatures has also allowed the identification of molecular subgroups with different prognoses or predictive treatment responses in ccRCC. Five large studies have classified ccRCC into subgroups based on transcriptomic data (Table 1). First, Brannon et al. found two subgroups with different biological signatures and prognoses (ccA and ccB) [117]. The KIRC analysis determined four clusters (m1-m4) [118] and Chen et al. identified three subgroups (CC-e.1, CC-e.2, CC-e.3) [119]. These three classifications overlap. Indeed, the m1 cluster of KIRC is associated with the ccA and CC-e.2 group, the m3 group with ccB and CC-e.3 while CC-e.1 is associated with m1 and m4. This allows us to divide them into three different clinical outcome groups: (1) good prognosis corresponding to cluster ccA, CC-e.2, and m1 associated with overexpression of angiogenic gene, mutations of PBMR1 and chromatin remodeling genes, (2) poor prognosis corresponding to cluster ccB, CC-e.3, m3 associated with mutation of cell cycle genes (CDKN2A), many hypoxia-related genes, chromatin modification genes (SETD2) and PI3K/AKT/mTOR pathway, (3) intermediate prognosis corresponding to CC-e.1, m2, and m4 clusters associated with mutations in the BAP1 gene and base-excision repair [16]. Beuselinck et al. performed a multiomics analysis that identified four molecular subgroups predicting response to sunitinib in ccRCC. The ccrcc1 (“c-myc-up”) and ccrcc4 (“c-myc-up and immune-up”) are characterized by the upregulation of MYC targets; ccrcc2 (“classical”) and ccrcc3 (“normal-like”) present a higher expression of the pro-angiogenic HIF-VEGF-VEGFR-pathway [120]. From this study, a reduced signature of 35 genes was developed to differentiate ccRCC patients according to these four groups. The ccrcc2 group displayed the highest expression of the pro-angiogenic HIF-VEGF-VEGFR-pathway (VEGF-A, VEGFR-1, VEGFR-2, and HIF2A), particularly in tumours with a bi-allelic *PBRM1* loss of function [115]. This transcriptomic profile was associated with better PFS and OS in patients receiving sunitinib. The ccrcc4 group, which had the poorest prognosis, was associated with high inflammation, decreased angiogenesis, and resistance to sunitinib and pazopanib [121,122]. Finally, in the phase 3 COMPARZ trial evaluating the efficacy of sunitinib and pazopanib in m-ccRCC, the study of 1500 genes identified four biological distinct clusters (Cluster 1–4) [35]. Angiogenesis gene expression (established on an existing signature [123]) was significantly different between clusters. High expression of angiogenesis genes was associated with better ORR (*p* = 0.03) but also better PFS and OS (HR 0.68; 95% CI: 0.52–0.90; *p* = 6.11 × 10^−3^; HR 0.68; 95% CI: 0.53–0.88; *p* = 2.49 × 10^−3^ respectively) compared to the Angio-low group. The role of tumour macrophage infiltration on OS was also analyzed and poorer survival was found in patients with high macrophage infiltration. From there, they determined the association between angiogenesis gene expression and macrophage infiltration. The Angio-high Macro-low group had the best survival outcome for OS (HR 3.12; 95% CI: 1.93–5.03; *p* = 2.91 × 10^−6^) and PFS (HR 2.27; 95% CI 1.51–3.42; *p* = 8.58 × 10^−5^). Cluster 4, which presented the shorter OS, was unsurprisingly enriched in Macro-high Angio-low [35].

The relevance of trancriptomic studies including angiogenesis genes has been evaluated in patients treated with immunotherapy. The recent studies exploring the predictive role of molecular signatures assessed three main axes: angiogenesis, effector T response, and myeloid profile. These combined signatures were first tested to predict the response to immunotherapy alone or combined with antiangiogenic drugs. IMmotion 150 is a phase 2 trial evaluating atezolizumab (anti PD-L1) alone or in combination with bevacizumab versus sunitinib [36]. The Angio-high signature (see Table 1) was associated with higher vascular density (determined by CD31 staining in IHC), a better objective response rate (46% vs. 9%), and a better PFS (HR 0.31; 95% CI, 0.18–0.55) compared to Angio-low signature in patients treated with sunitinib. However, ORR and PFS did not significantly differ according to angiogenic profile in patients treated with atezolizumab alone or combined with bevacizumab. When evaluated across treatment arms, no apparent difference in PFS was observed in the Angio-high subgroup between the three arms of treatment. In the Angio-low group, the PFS was improved when patients were treated with the combination of atezolizumab plus bevacizumab at 11.4 months vs. 3.7 months in the group sunitinib and 5.4 months in the group atezolizumab (HR 0.59; 95% CI, 0.35–0.98). These results were confirmed in the IMmotion 151 phase 3 trial [126]. Furthermore, in agreement with previous trials, the Angio-high signature was associated with a better prognosis whatever the treatment. Javelin 101 is a phase 3 trial evaluating axitinib in combination with avelumab versus sunitinib. Again, patients with high expression of angiogenesis genes had better PFS than patients with low expression of these genes when treated with sunitinib but not in the avelumab plus axitinib arm. In patients with a low expression of these genes, PFS was improved when patients were treated with avelumab/axitinib in comparison to sunitinib [127].

This molecular signature was also used in the Checkmate 214 biomarker study evaluating nivolumab-ipilimumab versus sunitinib [124]. In this study, the Angio score was not predictive of PFS or OS in patients treated with nivolumab plus ipilimumab but was associated with longer PFS in patients receiving sunitinib. This is in accordance with the results of the BIONIKK trial [128]. This phase 2 multicentre non-randomized trial used ccrcc molecular groups to assign treatments in first-line advanced or m-ccRCC. The ccrcc1 (poorly immunogenic) and ccrcc4 (highly immunogenic) groups with lower sensitivity to sunitinib were randomized between nivolumab (N) alone or in combination with ipilimumab (NI). The ccrcc2 group, with strong angiogenic expression and high immune infiltration, was randomized between NI and anti-VEGF TKI. The ORR in the NI group was 39% for ccrcc1 (poor immune environment) vs. 50% for ccrcc2 and 4, which are both highly immunogenic but present different angiogenic profiles. Another recent study used whole transcriptome sequencing (WTS) in m-ccRCC patients treated with several ICI (ref Jee et al.). Three molecular subtypes were determined based on *PBRM1* mutation frequency. Subtype 2, which had the highest level of PBMR1 mutation, was enriched in genes involved in angiogenesis and metabolic pathways and was associated with better OS (*p* = 0.0042) compared to the other groups [125].

All these studies highlight the important role of angiogenesis in response to anti-angiogenic TKIs. In patients with low expression of angiogenesis genes, combination therapies seem to be an interesting choice depending on the immune cell content. Angiogenic signature does not predict the response to immunotherapy, either alone or in combination with AA.

#### 4.2.4. DNA Levels: Somatic Mutations

Although the inactivation of *VHL* is a frequent event in the carcinogenesis of both sporadic and hereditary RCC, no consistent correlation with patient prognosis has been demonstrated [129,130,131,132]. The possibility that it could serve as a predictive marker for the efficacy of antiangiogenic targeting VEGF-A has also been studied. Two studies report a higher response rate in patients with loss of function variants of *VHL* or *VHL* mutations compared to those with wild-type *VHL* when treated with anti-VEGFR (sunitinib, sorafenib, axitinib) or bevacizumab [30,133]. However, *VHL* mutation has failed to show predictive value in patients on anti-VEGFR [105,134,135]. The place of *VHL* mutations to predict response to immunotherapy has not been reported.

Somatic mutations of *PBRM1* seem to be predictive of a response to TKI. Indeed, a whole-exome sequencing was performed in two groups of patients treated with sunitinib or pazopanib. The analysis was performed on an extreme responder group (CR and PR > three years) and a primary resistant group (progression within the first three months). *PBRM1* mutation was significantly more common in extreme responders than in refractory patients (*p* = 0.01) [136]. Another study performed comprehensive genomic profiling by NGS on 31 cytoreductive nephrectomies in patients treated with various anti-angiogenic TKIs. *PBRM1* mutation was more frequent in patients who received prolonged treatment with anti-VEGF-A therapy [137]. *PBRM1* mutation has also shown interest as a predictive biomarker of response to immunotherapy. Miao et al. performed a whole-exome sequencing on tissue samples from 35 patients with RCC treated with nivolumab. Enrichment of *PBMR1* loss-of-function mutations at the tumour level was associated with increased clinical benefit (9/11 vs. 3/13). This analysis was validated in a cohort of 63 patients treated with ICI alone or ICI and anti-CTLA4 and found an association with clinical benefit (17/27 vs. 4/19) [37]. In another study including 189 patients with advanced ccRCC treated with nivolumab, *PBRM1* mutation was associated with a better response and improved PFS. This association was not found in patients treated with everolimus [138]. *PBRM1* mutation was also associated with improved OS in patients receiving nivolumab [139]. However, in recent studies including patients treated with anti-angiogenic TKIs alone or in combination with anti-PD-1, *PBRM1* mutation is still associated with a response to TKIs alone but not when combined to anti-PD-1. Mutation of *PBRM1* would be associated with a less immunogenic TME and a decrease in the IFN-γ-JAK2-STAT1 signaling pathway, thus responsible for a decrease in the efficacy of ICI compared to TKI [140]. Another study characterizing the TME using gene expression signatures described two clusters of patients: one called the “inflamed” subtype, enriched for immune modulatory cells, and specifically associated with *BAP1* mutations, and a second group characterized as “noninflamed”, enriched for angiogenesis, dendritic cells, and mast cells [141]. In this study, *PBRM1* mutations were found to be homogeneously distributed across both groups.

Based on the sequencing of 101 ccRCC tumours (TRACERx consortium), Turajlic et al. showed *VHL* mutations to be a consistent clonal event found in main tumours (77/106), which is consistent with the low impact of the mutational status of *VHL* to guide therapies [37]. They described three types of tumors presenting different growth patterns: *VHL* inactivation and chromosome 3p loss had an indolent growth course in comparison to those with *PBRM1* mutations, while tumours mutated for both *BAP1* and *PBRM1* presented rapid progression. These findings suggest that evolutionary classification could be helpful to optimize treatment options for ccRCC patients [37]. However, 2 and 7 biopsies are needed to capture 50% and 75% of the driver events, respectively, which is a limit to using TRACERx genetics to stratify patients into treatment algorithms.

## 5. Discussion

This review highlights the important role of angiogenesis not only in tumour development but also in the resistance to treatment, mainly through its interaction with hypoxia and immunity. Angiogenesis markers appear to be useful in predicting response to anti-VEGFR TKIs. Many studies have assessed circulating biomarkers of angiogenesis, particularly those related to the VEGF-A pathway in patients treated with sunitinib but few data are available in patients treated with ICI alone or combined with TKI. High VEGF-A level at baseline was consistently associated with worst outcomes in patients treated either with TKI or with ICI. Soluble VEGFR-2 blood level systematically decreases when patients are treated with TKI but only some studies report an association with the response to treatment. The study of other blood markers related to angiogenesis and their monitoring shows discordant results. Patient cohorts differ between studies leading to differences in baseline biomarker levels and thus in the threshold used to compare the outcomes of the patients. Standardization in the choice of threshold values is necessary to be able to validate and use blood biomarkers in clinical practice. Inconsistent results could also be partly explained by the diversity of angiogenic pathways that can be activated in the tumour or the involvement of other pathways. Concerning the tumour tissue biomarkers, the most advanced and reproducible results were obtained with the transcriptomic data thanks to currently available technologies. Several molecular signatures have been tested in patients treated with TKI or ICI alone or combined. Finally, several studies have shown evidence that upregulation of a set of angiogenesis-related genes seemed to help predict a better response to anti-VEGF therapy, whether alone or combined. Patients with low angiogenic tumour are less likely to respond to anti-VEGF therapy alone and may benefit more from strategies involving ICI. The study of other pathways than angiogenesis is needed to determine which patients among low angiogenic ones will benefit from the ICI TKI combination or would need other therapeutic options like the anti-HIF-2α belzutifan.

Both blood biomarkers and molecular signatures present strengths and weaknesses for use in clinical practice. Transcriptomic analysis has the advantage over blood biomarkers of targeting complementary pathways. However, this technique is expensive and tissue biopsy is required which limits its use to following the patients. Moreover, the accuracy of the result is limited by tumour heterogeneity. Blood biomarkers have the advantage of reflecting both primary tumour and metastatic lesions, but caution must be taken for their interpretation as it may also reflect the systemic reaction to treatment [142,143]. The main strength of blood biomarkers is their accessibility with a simple blood sampling allowing longitudinal follow-up. The value of monitoring over time has yet to be determined but could allow early detection of resistance or help to sequence combined therapies. For several years, the development of multiplexing technologies has allowed the quantification of many cytokines in a small blood volume. Thanks to that, the combination of blood biomarkers providing a circulating angiogenic profile could be tested. Combining angiogenesis and immune-related blood biomarkers would also allow us to have a more complete overview of the TME, as is the case for transcriptomic data.

## 6. Conclusions

The search for angiogenesis-related biomarkers is informative to guide therapeutic choice but also to get insight into the role of angiogenesis in resistance mechanisms. It is necessary to better understand mechanisms underlying resistance to treatments to determine potential new therapeutic targets or combinations. Ongoing studies are focusing on new therapeutic targets such as HIF combined or not with immunotherapy or even on treatment escalation with triple therapy trials (double immunotherapy-AA). Combinations need to be optimized concerning the timing and the dose. Therapies that can be used as subsequent-line treatment also need to be defined. In all these trials, the place of related-angiogenesis biomarkers should be assessed and combined with markers of other pathways. Finding biomarkers of interest could not only help clinicians in the choice of these different treatments but also allow de-escalation of therapy by proposing less burdensome treatments in patients with the best chance of responding. 

## Figures and Tables

**Figure 1 cancers-14-06167-f001:**
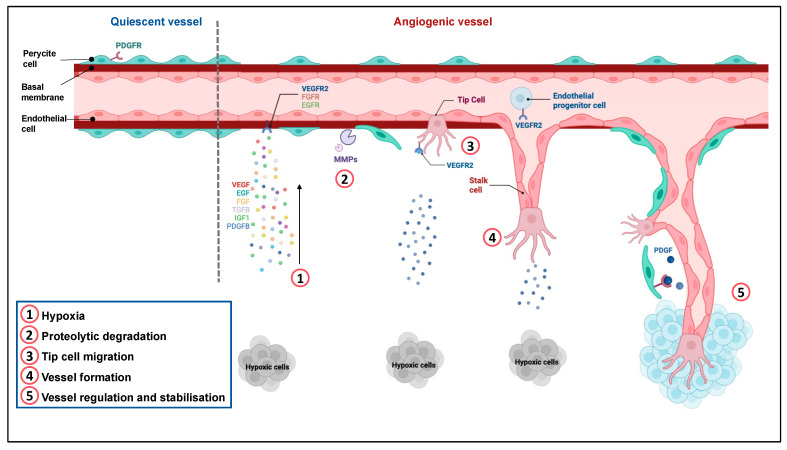
Regulation of angiogenesis. Quiescent ECs form a thin layer of single flat cells that line the interior surface of blood vessels and lymphatic vessels. These cells are interconnected by junctional molecules. EC monolayer is covered by pericytes, which control EC proliferation, release cell-survival signals and produce the basement membrane. ①. Hypoxia induces the secretion of pro-angiogenic factors. ②. This secretion leads to pericyte detachment, basement membrane degradation by metalloproteases and loss of EC junctions via VEGFR-2 activation. ③. One EC called tip cell is selected to guide elongation of the vessel towards proangiogenic signals. Remodeling of the existing matrix allows the migration of ECs. ④. Stalk cells proliferate and elongate to form a new vessel. ⑤. Following Dll4-Notch signaling and PDGF release, EC resume their quiescent state, the vessel is stabilized by recruitment of pericytes via PDGFR and deposition of a basement membrane. Adapted from “Tumor vascularization”, by BioRender.com (2022). Retrieved from https://app.biorender.com/biorender-templates (accessed on 4 December 2022).

**Figure 2 cancers-14-06167-f002:**
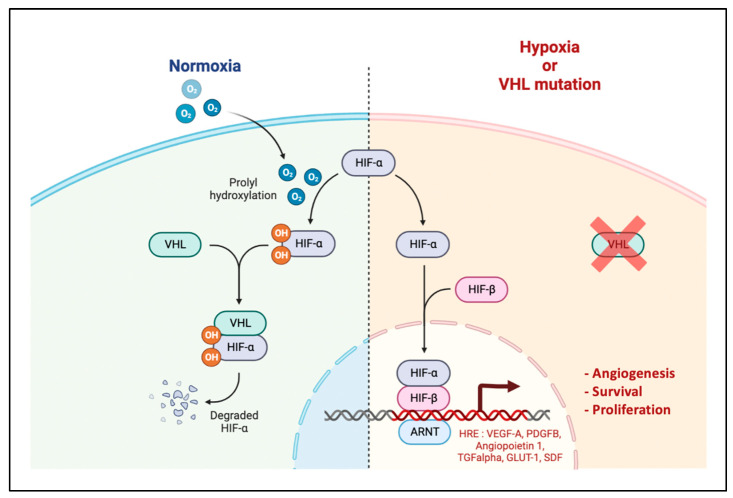
HIF Pathway. Under normoxic conditions, HIF-α is hydroxylated by prolyl hydroxylases (PHD) and then recognized by the VHL protein. Once VHL is bound to HIF-α, this leads to its ubiquitination and degradation by the proteasome. Under hypoxic conditions, HIF-α is not hydroxylized, accumulates in the cytosol, translocates to the nucleus and heterodimerizes with its subunit HIF-β. This leads to the transcription of genes with a hypoxia responsive element (HRE) in their promoter, responsible for cellular adaptation to hypoxia such as angiogenesis, survival, glucose metabolism and proliferation. Adapted from “HIF signalling”, by BioRender.com (2022). Retrieved from https://app.biorender.com/biorender-templates (accessed on 4 December 2022).

**Figure 3 cancers-14-06167-f003:**
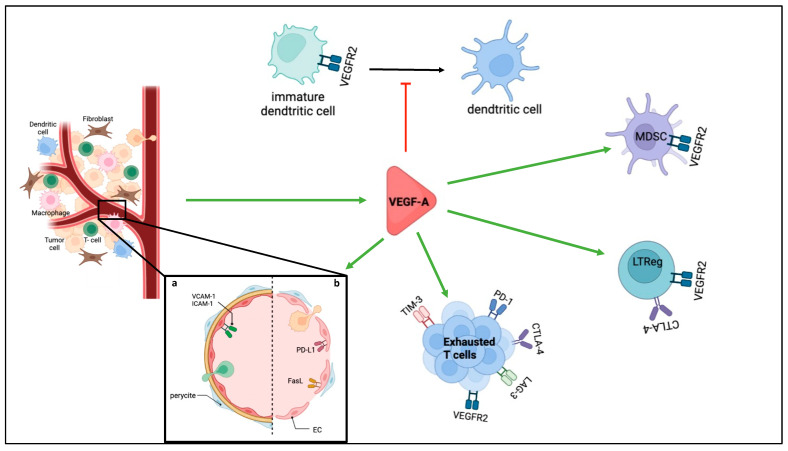
Immunosuppression role of VEGF-A. Different immunosuppressive roles of angiogenesis are presented, either through the immunosuppressive functions of proangiogenic factors such as VEGF-A or through the immunosuppressive functions of the tumour blood vessels. a. illustrates the structure of normal blood vessels when activated with TNF-α which favours infiltration of immune cells, b. illustrates tumour blood vessels with altered structure and examples of immunosuppressive functions by expression of immunosuppressive molecules.

**Figure 4 cancers-14-06167-f004:**
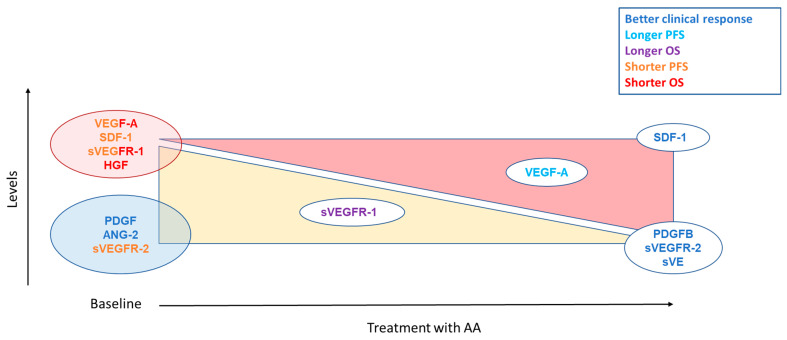
Summary of angiogenesis markers assessed as predictive factors for TKIs. The yellow triangle corresponds to a decrease of the markers on treatment. The red triangle indicates an increase in markers on treatment. For biomarkers placed in the middle of the triangle, it indicates that a lower decrease or a lower increase is associated with the clinical response corresponding to their color.

**Table 1 cancers-14-06167-t001:** Summary of transcriptomic data as prognostic or predictive factors.

Transcriptomic Classification	Angiogenic Profile Based on mRNA Expression of:	Angio High Group	Treatment	Pronostic/ Predictive	Refs.
*ccA* *ccB*	FLT4, FLT1, VEGFB, ENG, KDR, BAI1	ccA	ø	Good Poor	[117]
*M1* *M2* *M3* *M4*	ø	M1	ø	Good Intermediate Poor intermediate	[118]
*CC-e.1* *CC-e.2* *CC-e.3*	ø	CC-e.2	ø	Intermediate Good Poor	[119]
*ccrcc1* *ccrcc2* *ccrcc3* *ccrcc4*	VEGF-A, FLT1, KDR, and HIF2A	ccrcc3 >ccrcc2	sunitinib	Shorter PFS and OS Shorter PFS and OS	[115,120]
*Cluster 1* *Cluster 2* *Cluster 3* *Cluster 4*	CDH5, ELTD1, CLEC14A, LDB2, ECSCR, MYCT1, RHOJ, VWF, TIE1, KDR, ESAM, CD93, PTPRB, GPR116, SPARCL1, EMCN, ROBO4, ENG, TEK, S1PR1	Cluster 3	Pazopanib vs. sunitinib	Angio-high Signature: -Better ORR -Longer PFS and OS Angio-high Macro-low group: -Longer PFS and OS Cluster 4: Shorter OS	[35]
*ø*	VEGF-A, KDR, ESM1, PECAM1, ANGPTL4, CD34	N/A	NI Or sunitinib	Angio high -Sunitinib: longer PFS -NI: NS	[124]
*1* *2* *3*	Angio signature not detailed	Subtype 2	Nivolumab Or NI	Longer OS	[125]
*ø*	VEGF-A, PECAM1, ANGPLT4, ESM1, FLT1, CD34, KDR	N/A	AB or A or sunitinib	Angio high - Better prognosis -Sunitinib: Better ORR. Longer PFS -AB and B: NS Angio low -AB: Longer PFS	[36,126]
*ø*	VEGF-A, KDR, ESM1, PECAM1, ANGPTL4, CD34	N/A	AA versus sunitinib	Angio high -Sunitinib: Longer PFS -AA: NS Angio low -AA: Longer PFS	[127]

NI: Nivolumab + ipilimumab. AB: Atezolizumab + bevacizumab. A: Atezolizumab. AA: Avelumab + axitinib N/A: not applicable. NS: not significant.

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
