# Peer review of "Renal Carcinoma and Angiogenesis: Therapeutic Target and Biomarkers of Response in Current Therapies"

_cancers, 2022, doi:10.3390/cancers14246167_

Round 1

Reviewer 1 Report

The authors present a work about biomarkers related woth angiogenesis in response in RCC. The manuscript is well written and prove concise information but some changes are needed to be adressed by authors before publication:

English need a slight review across manuscript

I suggest adding a "Discussion" paragraph in which the Authors summarize the results of their literature's revision, providing their perspectives on this topic.

I suggest adding a “Methods” paragraph in which the Authors explain how they performed the research and how the results are displayed, considering the large amount of data reported.

Resistance mechanisms related with antiangiogenic treatment are lacking in manuscript. The authors should add it to improve manuscript quality. I suggest add following references (doi: 10.3390/cancers13235981, https://doi.org/10.1186/s13046-021-01961-3)

Author Response

The authors present a work about biomarkers related woth angiogenesis in response in RCC. The manuscript is well written and prove concise information, but some changes are needed to be adressed by authors before publication:
We would like to thank the reviewer for these comments.

Point 1: English need a slight review across manuscript

Response 1: The whole text has been revised by a native english speaker.

Point 2: I suggest adding a "Discussion" paragraph in which the Authors summarize the results of their literature's revision, providing their perspectives on this topic.

Response 2: We have added a discussion and perspectives section in the conclusion to resume the most important data detailed in the review and to provide an insight into the different ways of using these tools in the future, in red in the revised manuscript.

Point 3: I suggest adding a “Methods” paragraph in which the Authors explain how they performed the research and how the results are displayed, considering the large amount of data reported.

Response 3: A « Methods » paragraph has been added in the review, in red in the revised manuscript.

Resistance mechanisms related with antiangiogenic treatment are lacking in manuscript. The authors should add it to improve manuscript quality. I suggest add following references (doi: 10.3390/cancers13235981, https://doi.org/10.1186/s13046-021-01961-3) :

Response 4: A new paragraph detailing the place of angiogenesis among the mechanisms of resistance to TKI and immunotherapies has been included at the end of Part 3. Angiogenesis in renal cancer. Titles have been changed to better include this new part. The reference advised by the reviewer is cited here for details on the mechanisms of resistance other than angiogenesis. All these changes are in red in the revised manuscript. 

Reviewer 2 Report

The Review article entitled “Renal carcinoma and angiogenesis: Therapeutic target and biomarkers of response in current therapies” discussed the role of Angiogenesis in renal cancer, Angiogenesis blood biomarkers for cancer, etc. The article has many grammatical and sentence errors, and the language organization needs to be improved. For these reasons, I conclude that the paper should undergo minor revision

1.      The introduction is good but very general in nature. Authors need to provide more insight into about the role of angiogenesis cancer development with the importance of this review in accordance with the importance of angiogenesis biomarkers in cancer diagnosis.

2.      Authors need to improve the references by citing recent references like

https://doi.org/10.3390/cancers14061406

https://doi.org/10.1038/s41419-022-04898-3

https://doi.org/10.1038/s41420-022-00865-1

https://doi.org/10.3390/cancers14030644

3.      Figure 1 readability can be improved by working on the color used for the text.

4.      Typographical errors can be avoided. The language and grammar used throughout the manuscript need to be improved. 

5.      Conclusion needs to improve by providing a future perspective of angiogenesis biomarkers in cancer diagnosis and treatment to improve to prognosis. 

Author Response

The Review article entitled “Renal carcinoma and angiogenesis: Therapeutic target and biomarkers of response in current therapies” discussed the role of Angiogenesis in renal cancer, Angiogenesis blood biomarkers for cancer, etc. The article has many grammatical and sentence errors, and the language organization needs to be improved. For these reasons, I conclude that the paper should undergo minor revision

We would like to thank the reviewer for these comments.

Point 1. The introduction is good but very general in nature. Authors need to provide more insight into about the role of angiogenesis cancer development with the importance of this review in accordance with the importance of angiogenesis biomarkers in cancer diagnosis.

Response 1: the introduction has been modified to introduce the role of angiogenesis in tumour development and response to treatment, in red in the revised manuscript.

2. Authors need to improve the references by citing recent references like
https://doi.org/10.3390/cancers14061406 :
https://doi.org/10.1038/s41419-022-04898-3 :
https://doi.org/10.1038/s41420-022-00865-1
https://doi.org/10.3390/cancers14030644

Response 2: Papers dealing with lncRNA have been cited in a new part “3.3. Role of angiogenesis in resistance to treatment” to illustrate the role of lncRNA in resistance to treatment, in red in the revised manuscript. The review of Kasherman et al. explaining the rational of combining anti-angiogenic drugs and immunotherapies is cited in the introduction part.

3. Figure 1 readability can be improved by working on the color used for the text.

Response 3: Figure 1 has been modified for clarity.

4. Typographical errors can be avoided. The language and grammar used throughout the manuscript need to be improved.

Response 4: The manuscript has been entirely revised.

5. Conclusion needs to improve by providing a future perspective of angiogenesis biomarkers in cancer diagnosis and treatment to improve to prognosis.

Response 5: We have added a discussion and perspectives section in the conclusion, in red in the revised manuscript, to resume the most important data detailed in the review and to provide an insight into the different ways of using these tools in the future.

Round 2

Reviewer 1 Report

The authors adressed all comments